# In Vitro Anti-Inflammatory and Regenerative Effects of Autologous Conditioned Serum from Dogs with Osteoarthritis

**DOI:** 10.3390/ani12192717

**Published:** 2022-10-10

**Authors:** Sirikul Soontararak, Piyathida Ardaum, Napaporn Senarat, Sarawut Yangtara, Chalermpol Lekcharoensuk, Iyarath Putchong, Narudee Kashemsant, Monchanok Vijarnsorn, Lyndah Chow, Steven Dow, Porntippa Lekcharoensuk

**Affiliations:** 1Department of Companion Animal Clinical Sciences, Faculty of Veterinary Medicine, Kasetsart University, Bangkok 10900, Thailand; sarawut.y@ku.th (S.Y.); fvetcpl@ku.ac.th (C.L.); monchanok.vi@ku.th (M.V.); 2Kasetsart University Veterinary Teaching Hospital, Faculty of Veterinary Medicine, Kasetsart University, Bangkok 10900, Thailand; piyathida.ar@ku.th (P.A.); napaporn.se@ku.th (N.S.); 3Department of Microbiology and Immunology, Faculty of Veterinary Medicine, Kasetsart University, Bangkok 10900, Thailand; iyarath.p@ku.th; 4Department of Physiology, Faculty of Veterinary Medicine, Kasetsart University, Bangkok 10900, Thailand; nkashemsant@gmail.com; 5Center for Immune and Regenerative Medicine, Department of Clinical Sciences, College of Veterinary Medicine and Biomedical Sciences, Colorado State University, Fort Collins, CO 80523, USA; lyndah.chow@colostate.edu (L.C.); steven.dow@colostate.edu (S.D.)

**Keywords:** autologous conditioned serum, canine, osteoarthritis, bioregenerative therapy, hemoderivatives, Interleukin 1 receptor antagonists, cytokines, growth factors, T cells suppression

## Abstract

**Simple Summary:**

Autologous conditioned serum (ACS) has benefits in managing osteoarthritis (OA) due to enriched mediators, including Interleukin-1 receptor antagonists (IL-1RA) and growth factors. Although ACS has become widely used in the clinic with favorable treatment outcomes in humans and animals, the study focusing on the therapeutic potential of ACS in dogs is limited. Therefore, this study investigated the therapeutic potential of ACS generated from dogs with OA, focusing on the potential anti-inflammatory and regenerative properties. We showed that ACS from OA dogs has essential mediators with immunomodulatory effects on reducing inflammation and promoting joint regeneration. We showed that ACS increased T cell suppression and FOXP3^+^ T cell expansion in vitro. In addition, ACS enhanced the proliferation of joint cells and promoted chondrocyte extracellular matrix gene expression. These actions are probably caused by the mediators in ACS, including transforming growth factor, vascular endothelial growth factor, and IL-1RA. This study provides insight into the benefits of an autologous bioactive substance derived from target users as a promising potential in regenerative strategies for OA management.

**Abstract:**

Osteoarthritis (OA) is mostly incurable and non-regenerative with long-term complications. Autologous conditioned serum (ACS), which is enriched in Interleukin 1 receptor antagonists (IL-1RA) and growth factors, could be an alternative treatment to accelerate the positive therapeutic effects. ACS is proposed to alleviate inflammation by blocking IL-1 receptors. However, to date, there is no report focusing on the cell-mediated anti-inflammation and regenerative effect caused by ACS, especially the ACS from patients. Therefore, this study aims to investigate the therapeutic potential of ACS generated from dogs with spontaneous OA, focusing on its promising anti-inflammatory and regenerative properties in vitro compared to the matched plasma. We found that ACS prepared from ten OA dogs contained significant concentrations of IL-1RA, vascular endothelial growth factor, and transforming growth factor beta, which are key cytokines in anti-inflammation and angiogenesis. Furthermore, we found that ACS suppressed T cell activity by reducing proliferation of effector T cells and simultaneously expanding numbers of immune suppressive FOXP3^+^ T cells. Lastly, we showed that ACS enhanced the proliferation of osteocytes and fibroblasts and promoted extracellular matrix gene expression in primary chondrocyte culture. Therefore, these studies indicate that ACS prepared from dogs with OA is active as an immunomodulatory and regenerative strategy for use in OA management.

## 1. Introduction

OA is a chronic progressive inflammation of the joint resulting in degeneration of hyaline cartilage, subchondral bone alteration, osteophytosis, and impaired joint function [1,2,3]. OA becomes a growing problem in aging humans and companion animals [4]. The disease is primarily an irreversible, incurable, non-regenerative condition with long-term complications despite receiving standard therapy typically including a combination of pain control medication, joint supplement, and physiotherapy [2,5,6]. An alternative intervention to alleviate joint inflammation is using cytokine inhibitors specifically the Interleukin-1 Receptor Antagonist (IL-1RA) [7,8].

Competitive binding at the IL-1 receptor by IL-1RA is considered the primary action inhibiting the joint’s inflammatory process [7,9]. Studies have shown that IL-1RA prevents IL-1 ligand activation resulting in reduced synthesis of mediators such as metalloproteinases (MMPs), cyclooxygenase-2 (COX2), prostaglandins, and nitrous oxide that propagate OA [1,9,10,11]. Furthermore, the high concentration of IL-1RA in joints is crucial to maintain joint homeostasis whereas low levels are observed in synovial tissues from the OA patient [12]. Thus, the alternative sustainable treatment that potentially accelerates therapeutic effects is the use of autologous condition serum (ACS) which is enriched in IL-1RA and growth factors. These mediators in ACS are suggested to play a role in delayed joint destruction and promoted regeneration. However, the ACS effect on the cell-mediated anti-inflammation and regeneration have not been reported. Thus in this work, we are specifically focusing on ACS from OA dogs which are the target user of the ACS product, and aim to investigate the mechanism by which the ACS could enhance therapeutic efficacy.

ACS is generated by in vitro incubating the whole blood from patients in an IRAP tube containing pre-treated glass beads for 6–24 h. The leukocytes, mainly the mononuclear cells, are activated to enrich immunomodulatory mediators, including growth factors and anti-inflammatory cytokines [13]. Several soluble growth factors in ACS are considered regenerative mediators for joint healing, including Transforming Growth Factor Beta (TGF-β), Vascular Endothelial Growth Factor (VEGF), and Fibroblast Growth Factor (FGF) [8,13,14,15]. Compared to the plasma, the inflammatory mediators in ACS are unchanged [16,17]. Thus, ACS is considered a safe and practical biological product, with the promising therapeutic effect on OA. The articular injection with ACS has become widely used in clinic with favorable treatment outcome in humans [8,18,19] and companion animals [7,20,21]. Noteworthy, the secretory mediators vary among the species and individuals [8,15,22], which is a common issue observed in hemoderivative products. ACS studies in equines showed the significant production of several mediators including IL-10, IL-1RA, TGF-β, and VEGF, corresponding to humans. To date, there is only one study reporting the component of ACS from healthy dogs [15]. Thus, the research focusing on the mediators in ACS from naturally occurring OA dogs is limited.

Using autologous bioactive agents enriched in anti-inflammatory mediators and growth factors for tissue regeneration has become a novel insight for OA management [11,23]. Apart from the IL-1 receptor modulation, several recent studies suggest that T cell-mediated response plays a significant role in the aggravation of OA, including an increase in activated T cells in synovial fluid and synovial membrane [24,25]. As a result, the immune mediated degeneration of the joint prevents the healing process. Thus, reduction in T cell activity is considered a crucial pathway for suppressing the inflammation while stimulating joint tissue regeneration, as we have seen in the mesenchymal stem cell treatment [26,27,28]. Although the clinical application of ACS has demonstrated a favorable result in limiting inflammation and pain [19,21,22,29], little is known regarding the ACS effect on T cell activity in dogs with spontaneous occurring OA whose immune response is primed and altered [30].

Therefore, this study aims to comprehensively investigate the potential therapeutic property of ACS obtained from spontaneous OA dogs. Specifically, we evaluated the T cell proliferation suppression, anti-inflammatory cytokines, and growth factors that potentially play a role in immunomodulatory effects and joint regeneration. In addition, we also assessed the regenerative effect of ACS on joint cell proliferation and extracellular matrix gene expression in chondrocytes in vitro.

## 2. Materials and Methods

### 2.1. Study Populations

The study was conducted at the Kasetsart Veterinary Teaching Hospital (KU-VTH). The study population was client-owned dogs presented to KU-VTH for evaluation and treatment of lameness. The study was approved by the Institutional Animal Care and Use Committee (IACUC) at Kasetsart University (ID#ACKU64-VET-024, approved on 21 March 2021). Dog owners were informed regarding the study protocol, and consent was obtained before enrollment in the study. A total of 10 dogs diagnosed with osteoarthritis were recruited for the study. The sample size was calculated using the key results from the previous publications in dogs [15] and other species [8,13] as well as the preliminary data. Moreover, the sample size was justified to achieve the power of the test based on the 3Rs framework and resources.

### 2.2. Osteoarthritis Dogs

Ten dogs with chronic osteoarthritis (five males and five females) with persistent signs of joint inflammation, including lameness, pain, joint swelling, and reluctance to move, for a minimum of 3 weeks were recruited into the study. All study dogs had general physical and orthopedic examinations, radiograph, complete blood count (CBC), and serum chemistry profiles to confirm a diagnosis of osteoarthritis and ruled out neuromuscular disorders. Most animals had previously undergone a standard intervention with poor response to the non-steroid anti-inflammatory drugs (NSAIDs) or physiotherapy. In addition, all dogs had no recent orthopedic surgery history, did not receive immunosuppressive medications, and were free from other diseases causing systemic illness, including pancreatic insufficiency as well as hepatic, metabolic, parasitic, and renal diseases.

### 2.3. ACS Sample Collection

Dogs were refrained from eating for 8 h prior to blood collection. They were allowed free access to the water until 6 h before sedation. Dogs were sedated with propofol (5 mg/kg, IV) and positioned in left lateral recumbency. The area at the right jugular vein was prepared for aseptic venipuncture. Fifty milliliters of venous blood sample was collected using a 20-gauge catheter and divided 10 mL each into three Orthokine^®^ Vet Irap tubes (Orthogen Veterinary GmbH, DUS, DE) and two tubes containing acid-citrate-dextrose (ACD-A) (BD, NJ, USA) for its sham plasma (SP) and T cell proliferation assay. Orthokine^®^ Vet Irap tubes contain pretreated glass beads which activate immune cells. ACS tubes and sham plasma tubes were processed within 30 min after collection by incubating in an incubator at 37 °C for 7 h as recommended by the manufacturer [15]. Then, the processed samples were centrifuged at 2000× *g* for 20 min at room temperature. ACS was collected and filtered using a 0.2 µM syringe filter. Two hundred microliters of ACS were aliquoted into 250 µL tube. ACS samples and serum were stored at −20 °C. To avoid the increased mediator production from the clot and platelet degranulation, the plasma was selected rather than serum [31]. The sham plasma (SP) was the whole blood from the same OA group collected in the ACD-A tube and was processed similarly to the ACS. All ACS properties were compared to their autologous sham plasmas since the plasma might contain cytokines and mediators to induce the bioactivity per se [32].

### 2.4. T Cell Proliferation Assay

Canine peripheral blood mononuclear cells (PBMC) were prepared from ACD-A whole blood using Ficoll gradient (LSM; MP Biomedicals Inc., Santa Ana, CA, USA). PBMC were resuspended in culture medium (RPMI, 10% FBS, 1% anti-anti) and were plated into a 96-well plate at 500,000 cells/well. Two assay controls, Concanavalin (ConA) (Sigma-Aldrich, St. Louis, MO, USA) and no ConA groups, and two groups of experimental conditions (ACS or SP) were set in triplicates. Concanavalin A at 10 µg/mL was put into the proliferation control and experimental wells to activate the T cell proliferation [28]. Then, 50 µL of ACS or SP was added to the experimental groups. Subsequently, 10 µM EdU (5-ethynyl-2′-deoxyuridine) were added into all wells. The culture plate was incubated at 37 °C with 5% CO_2_ for 72 h.

To assess T cell proliferation, the cultured PBMC were labeled with anti-dog CD5-APC (Clone YKIX322.3; Thermo Fisher Scientific, Waltham, MA, USA) and anti-mouse/rat FOXP3-PE (Clone FJK-16S; Thermo Fisher Scientific). The isotype controls for flow cytometry gaiting were also stained using the same method. Then the cells were stained using Click-iT™ EdU Alexa Fluor™ 448 Assay Kit (Thermo Fisher Scientific), according to the manufacturer’s directions. The stained cells were analyzed on 50,000 events of CD5^+^ cells using flow cytometry (Cytoflex^®^, Beckman Coulter, CA, USA). Flow cytometry data were analyzed using FlowJo Software (Ashland, OR, USA). T cell proliferation was measured as the percent EdU positive within the gated CD5^+^ population of cells. The calculation of T cell proliferation suppression was modified from the previous study [33]. The data were calculated based on the suppression percentage normalized to their values with those from the assay control wells. For calculation, the suppression percentage of the assay control was assigned to the maximum value at 100% suppression (100% suppression for no ConA; 0% suppression for ConA). Flow cytometry gating area result using forward scatter (FSC), side scatter (SSC), histogram, and percentage calculation were demonstrated in the Results section.

### 2.5. ELISA for Cytokine Production

Cytokine levels in ACS and SP were assessed using canine-specific ELISAs. IL-1β, VEGF, IL-10, and TNF-α were measured using DuoSet canine-specific ELISA from R&D Systems Inc. (Minneapolis, MN, USA). IL-1RA and TGF-β1 ELISA kits were from MyBiosource, Inc. (San Diego, CA, USA). For IL-1RA and TGF-β1 quantification using ELISA, the ACS and SP samples were diluted 1:3 in phosphate buffer saline (PBS) according to the manufacturer’s recommendation. For other ELISAs, including IL-1β, IL-10, TNF-α, and VEGF, sample dilution was not necessary. All ELISAs were performed according to the manufacture recommended protocols and repeated for three independent rounds with three technical replicates.

### 2.6. Cell Proliferation Assay

To assess the cell proliferation supported by ACS, the colorimetric method (MTS assay) (CellTiter96^®^, Promega Corp., MI, USA) was performed. The canine fibroblast (A-72) and canine osteocyte (D-17) were obtained from ATCC (Manassas, VA, USA) and were cultured following the company guidelines. Briefly, the cells (osteocytes or fibroblasts) were plated into 96-well plates at the density of 10,000 cells/well with 150 µL of culture medium. The culture media were prepared as recommended by the manufacturer’s protocol. Then, the 50 µL of ACS or SP were added into the experimental wells and incubated for 48 h. The assay control was the cells fed with the recommended medium and the same incubating duration as the cells in the experimental wells. After 48 h, the MTS assay was performed according to manufacturer’s protocol. The sample optical density (OD) values was measured at 490 nm using a microplate reader. The proliferation index was calculated as the proliferation ratio normalized to the OD value of the assay control well, which was assigned as a value of 1. In other words, the ratio was calculated by dividing the OD of experimental wells by the OD of controls. Three independent experiments were performed for each cell line, and all conditions were set in triplicate.

### 2.7. Analysis of Chondrocyte Gene Expression

To assess the chondrocyte response supported by ACS, the expression of extracellular matrix (ECM) producing genes was analyzed. Briefly, the canine chondrocytes were isolated from the articular cartilage following the previously established protocols [34]. In passage 3, the primary chondrocytes were plated at 50,000 cells/mm^2^ in a 24-well plate [34]. Canine primary chondrocytes were incubated with ACS or SP of five individuals for 24 h. Total RNA was extracted from culture using TRIZol reagent (Invitrogen, Carlsbad, CA, USA). The cDNA of each sample was constructed using a cDNA synthesis kit (ReverTra Ace -α-^®^TOYOBO, Co., Ltd., Osaka, Japan) following the manufacturer’s instructions and preserved at −30 °C until further used. To obtain the expression profiles, the targeted genes were amplified using primers specific to *Collagen type I, II, MMP-13, SOX9, Aggrecan* [34]. *GAPDH* was used as a housekeeping gene (Appendix A). The real-time PCR reaction was conducted using KAPA SYBR FAST qPCR kit (KAPA Biosystems, Woburn, MA, USA) according to the manufacturer’s instructions. Then CT values were generated using the Bio-Rad CFX96™ Real-Time PCR Detection System. The fold change in the gene expression level was calculated using the ∆∆CT method (2^−ΔΔCT^) [35] based on the data that were normalized to untreated control and housekeeping gene. The fold change in gene expression was compared between groups.

### 2.8. Statistical Analysis

Data were analyzed using Prism 9 software (GraphPad, San Diego, CA, USA). The normality of data was initially analyzed using the Shapiro–Wilk normality test. The normally distributed data were shown as mean ± standard deviation (SD). Data that were not normally distributed were reported as the median (range). Statistical differences between the two groups (ACS vs. SP) were evaluated using the two-tailed paired *t*-test for parametric data and the Mann–Whitney test for non-parametric data, as indicated in the text. The results from repeated experiments, including the MTS assay result, were normalized to baseline control before analysis. For gene expression analysis, the fold change in expression was compared between ACS and SP using paired *t*-test if the data were normally distributed. The log transform was performed if the data were not normally distributed. In all studies, the statistical significance was set at *p* value < 0.05.

## 3. Results

### 3.1. Demographic of Study Dogs

Ten dogs were recruited for the study according to the inclusion criteria stated in Methods. The demographic data, disease duration, and affected joint (s) of the OA dogs enrolled in the study are shown in Table 1. The breeds included Labrador Retriever (*n* = 3), Golden Retriever (*n* = 3), German Shepherd, Crossbreed, American Pitbull, and Siberian Husky. On average, the disease duration in the studied dogs was 5.8 weeks (median four weeks), with single or multiple affected joints.

### 3.2. ACS Enriched Mediators for Regenerative and Anti-Inflammation

The cytokine enrichment in response to ACS preparation was evaluated using ELISA. Our study found that ACS contained a significant level of the anti-inflammatory cytokine, IL-1RA, and growth factors; VEGF and TGF-β1 but the limited level of IL-1β (Figure 1). The level of IL-1RA in ACS group (816.1 ± 425.9 pg/mL) were significantly higher than SP group (436.9 ± 286.1 pg/mL; *p* = 0.0001). In addition, statistically significant difference of VEGF level in ACS was found (*p* = 0.0020; ACS (65.85 ± 33.60 pg/mL) vs. SP (20.23 ± 19.79 pg/mL). Interestingly, the ACS has a dramatically higher concentration (conc.) of TGF-β1 level compared to the sham plasma (*p* = 0.0014; ACS (4172 ± 2286 pg/mL) vs. SP (489.5 ± 520.4 pg/mL)). In contrast, there was a limited concentration of IL-1β (ACS; median 27.91 pg/mL (0–146.5 pg/mL) equivalent to sham plasma (median 43.69 pg/mL (0–300.6 pg/mL)). We found less production of IL-10 and no difference between groups (ACS; median 19.58 pg/mL (0–972.7 pg/mL), SP; median 28.10 pg/mL (0–1295 pg/mL)). TNF-α level was low in several samples and undetectable by the ELISA.

### 3.3. Impact of ACS on T Cell Proliferation Activity

To investigate whether ACS has anti-inflammatory effects on the immune response, T cell activity from the patient in response to ACS or SP was determined using matched samples. Concanavalin A (ConA) was added to stimulation controls and all experimental wells (ACS and SP). T cell proliferation induced by ConA showed scattered distribution of cells (Figure 2a; +ve Ctrl). Without ConA, the unstimulated cells (Figure 2b; -ve Ctrl (negative control)) accumulated at the bottom of the U-shape wells with unchanged morphology. In addition, we noticed increased cell clumping indicating no proliferation in the ACS treated group compared to the SP group (Figure 2c,d). Subsequently, these cells were immunostained and quantified using flow cytometry.

In flow cytometry analysis, T cell proliferation was assessed by calculating the percentage of EdU^+^ (Alexa Fluor™ 448) within the gated CD5^+^ population. Because the T cell activity may vary between individual dogs, T cell proliferation suppression was calculated as mentioned previously in Methods. Flow cytometry gating area and representative percentage calculation were demonstrated in Figure 3.

Notably, both ACS and SP groups showed an ability to inhibit T cell proliferation since the mean fluorescent intensity (MFI) of the EdU was significantly reduced compared to the stimulated positive cell control. The percentage suppression range was approximately 10–90% (Figure 4). Significantly increased suppression was observed in the ACS-treated T cells (*p* = 0.0301) compared to the SP group (Figure 4a). The percentage of T cell suppression by ACS and SP was approximately 62.24% ± 18.96 and 47.58% ± 29.26, respectively. However, the MFI ratio of the incorporated EdU DNA was lesser in the ACS group (MFI-ACS: 0.37 ± 0.19 a.u.; MFI- SP: 0.50 ± 0.36 a.u., *p* = 0.02; Figure 4b).

Interestingly, we found a significantly increased number of FOXP3^+^ cells in ACS-treated conditions (*p* = 0.0482). The populations of FOXP3^+^ T cells were 14.44% ± 8.52 and 6.10% ± 2.61 in the ACS and SP groups, respectively (Figure 4c). Greater MFI of FOXP3 were seen in the ACS group (MFI-ACS: 4757 ± 428.1 a.u.; MFI-SP: 3354 ± 171.9 a.u., *p* = 0.0099; Figure 4d). Therefore, ACS suppressed the T cell activation and induced the expansion of the FOXP3^+^ T cell population.

### 3.4. Promotion of Fibroblast and Osteocyte Proliferation by ACS

To evaluate the regenerative property of ACS to support cell growth, the canine fibroblast and osteocyte proliferations were investigated using the MTS assay. The proliferation index was calculated as the proliferation ratio normalized to the value from the cell culture feeding standard medium control, which was assigned as a value of 1. Treatment with ACS led to a significant level of cell proliferation in both fibroblasts (*p* = 0.0048) and osteocytes (*p* = 0.0189) compared to SP treatment group (Figure 5). The fibroblast proliferation ratios induced by ACS and SP were 1.022 ± 0.041 and 0.912 ± 0.095, respectively. ACS induced the greater osteocyte proliferation ratio than SP (ACS; 0.900 ± 0.037, SP; 0.780 ± 0.130).

### 3.5. ACS Effects on Cartilage Matrix mRNA Expression

Several previous studies have reported the effectiveness of autologous hemoderivative products in controlling cartilage degradation in humans [11,36] and animals [32,37]. Therefore, we further investigated whether the ACS could alter the expression of ECM producing genes in primary chondrocytes. The mRNA expression of protein representing the cartilage matrix was quantitated by RT-qPCR. The fold change in mRNA expression normalized to control using Δ∆CT was calculated as previously stated in Methods. ACS stimulated the greater significant fold change in Collagen type II expression (*p* = 0.0247) compared to SP (Figure 6a) in primary chondrocytes. For other genes, including Collagen type I, Matrix metallopeptidase 13 (MMP-13), Aggrecan, and SRY-Box Transcription Factor 9 (SOX9), the fold changes in expression were not significantly different between the two groups.

## 4. Discussion

OA dogs experience long-term inflammation, pain, and irreversible destruction of the joints, as with other degenerative diseases. One of the promising non-invasive alternative treatments that can potentially reduce inflammation and accelerate regenerative effects on the joint is ACS [8,20]. In this work, we studied a group of dogs with the naturally occurring OA with a single and also multiple joint presentations. With this chronic illness, the patient’s immune response may be altered since the monocytes believed to direct the type of cytokine production may be primed and affect the ACS components [20]. Thus, our results would represent the actual condition of the target users, in which their hemoderivatives may be altered from the healthy candidates. In this study, we first reported a significant increase in TGF-β1 and VEGF in ACS of the OA dogs compared to the sham plasma control, similar to the previous report in human OA [16]. These growth factors are believed to play a role in joint homeostasis [25] and vascular regeneration [8,38], which are essential for joint and tendon repairs [36,39]. However, previous reports investigating ACS in normal dogs did not find the differences in these cytokines when compared to the normal serum [15,20].

In a placebo controlled clinical trial, ACS therapy reduced joint inflammation and pain, while recovering the joint function [18,22]. In addition, it has been successfully applied in the clinic. However, the effect of ACS on the autologous immune cells that mediate the anti-inflammation is not thoroughly studied. Our results showed that ACS could suppress T cell activity by reducing T cell proliferation and increasing the FOXP3^+^ T cell population (Figure 4). Although IL-10 is generally considered to play a role in anti-inflammation in OA [40], we did not observe the difference of this cytokine between the ACS and sham plasma control. Therefore, the T cell suppression is most likely caused by other mediators, possibly TGF-β, since the previous studies have reported the role of TGF-β in inducing the regulatory T cell expansion in the joint microenvironment [2,41]. This novel finding supports an advantage of ACS in immunomodulation to alleviate joint inflammation.

Previous studies have suggested that the FOXP3 expression was transient by in vitro induction and the activated, nonsuppressive T cells also expressed this marker [42]. From our results, ACS induced the increased number of FOXP3^+^ T cells which is more likely the regulatory T cells than the activated, nonsuppressive T cells because they suppressed the T cell proliferation. Thus, T cell-mediated response by ACS possibly helps to prolong the anti-inflammatory effects. Indeed, it has been shown that the concentration of the soluble mediators decreased rapidly within a few days after joint injection [17,32,43].

Apart from controlled inflammation, the crucial therapeutic strategy for OA is to promote local repair and stimulate joint tissue. We reported that ACS significantly promoted the proliferation of osteocytes, fibroblasts (Figure 5), and a member of the cartilage matrix components (Figure 6). This is most likely facilitated by the soluble bioactive mediators in ACS, especially TGF-β1 and VEGF, since a significant level of these cytokines presented in the ACS. TGF-β, FGF, IGF, and VEGF were previously reported to play a role in chondroprotection, cell proliferation, osteochondrogenic differentiation, and matrix production [36,44,45]. Moreover, we found the increased IL-1RA in the ACS which is necessary for cartilage homeostasis. In OA, reduction in IL-1RA over time leads to the loss the protective function in the joint [46]. Competitive binding of IL-1/IL-1RA potentially delays the progression of joint degeneration as reported previously [18,47]. Therefore, our result suggests that ACS therapy can be used for joint tissue regeneration and deceleration of the OA progression.

In addition, the immune cells and humoral mediators in OA patients are stimulated under chronic inflammation [11,30]. The response of leukocytes induced by the bead system in ACS may yield different product components. Thus, our study focused on comparing the ACS to the matched plasma control to normalize the bioactive mediators exerting by the individual plasma. These bioactive mediators could affect tissue regeneration and modulate immune responses [48,49,50]. Moreover, our result showed that the level of IL-1RA was consistently high in ACS dogs (Figure 1), corresponding to the previous studies in the healthy dogs [15]. In fact, IL-1RA is considered as a crucial mediator for blocking the inflammation induction in joints [11,13]. Previous studies in human and healthy dogs suggest that the effective ratio of IL-1RA:IL-1β for treating OA is 10–1000 [8,15,18]. We found that the estimated median of IL-1RA:IL-1β ratio is 27.23, which was sufficient for the therapeutic use.

Our results showed higher IL-1RA levels in ACS than in plasma. However, in the previous study in healthy dogs [15], the IL-1RA concentration was substantially higher compared to the present study. This is most likely caused by inter-animal variability and different health status. It is also possible that white blood cell activity from the OA dogs may have altered during the ACS preparation, resulting in the changes in cytokine production level. Thus, a further study on comparing ACS from healthy and disease populations would be the point of interest. Additionally, our results confirmed that the inflammatory cytokine levels in ACS from OA patients, including IL-1β and TNF-α, were not different from unconditioned plasma, similar to the previous report in equines OA and healthy dogs [13,15]. However, the different test kits and assays used in the studies might give the distinctive results. With this regard, we noted that the cytokine levels reported in the present study were from the canine-specific assays. Furthermore, inconsistent treatment outcomes were occasionally observed in clinical trials [22], suggesting a multifactorial effect of ACS [8]. This variation is also found in other hemoderivative products, including platelet-rich plasma (PrP) and concentrated protein [37].

Our study is a proof of the concept of the immunomodulatory and regenerative effects of the ACS generated from patient’s blood who is the target ACS user. However, comparison of biological mediators and effects exerting by ACS acquired from healthy and OA dogs should be further elucidated. This type of the study is a limitation in the clinical setting. Moreover, the study population retrieved from the clinic also possessed some variations, including breed, age, degree of inflammation, and affected site. The future study would consider adding a more diverse study population as well as healthy dog controls to fully emphasize the properties of ACS.

Moreover, to obtain an adequate volume of ACS, we intentionally selected only medium to large breed dogs which could tolerate the loss of substantial high blood volume. Thus, our study preferentially represented medium to large breed dogs. In addition, our in vitro study suggests the benefit of ACS in regenerative medicine. To verify the in vitro findings, a further study on the ACS effect in a controlled clinical trial in OA dogs would be the subsequent investigation.

Taken together, our study revealed the comprehensive in vitro investigation of therapeutic properties of ACS derived from the clinical OA patients. Our results provide evidence on the application of ACS as a promising supportive therapy in OA dogs according to its immunomodulatory and regenerative properties.

## 5. Conclusions

Intra-articular treatment with ACS is one of the non-surgical interventions that positively promote the success of OA management. Our results revealed that the ACS from the OA dogs contains essential mediators with the immunomodulatory effects on reducing inflammation, promoting joint cell proliferation and enhancing ECM gene expression by chondrocytes in vitro. This study provides insight into the benefits of an autologous bioactive substance derived from target users as a promising regenerative strategy for OA management.

## Figures and Tables

**Figure 1 animals-12-02717-f001:**
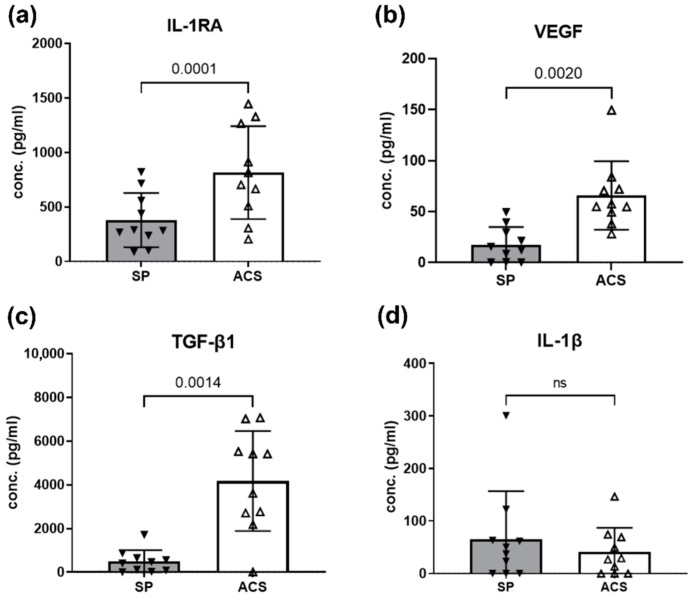
Growth factor and anti-inflammatory cytokine levels of ACS and SP derived from dogs with spontaneous OA. Bar graphs show the cytokine components in response to ACS as evaluated by ELISA. The levels of (**a**) IL-1RA, (**b**) VEGF, and (**c**) TGF-β1 in the ACS group were significantly higher than in unprocessed plasmas. *P* values were reported as indicated. There was a limited concentration of (**d**) IL-1β and IL-10 (graph not shown; data in Results). No significant difference of IL-1β level was observed in both groups. TNF-α concentration was below the detectable range of the kits. Filled and blank triangles represent the mean results obtained from 10 OA dogs with three independent replications and technical triplicates in each round. Error bars depict means with SDs in all panels. For all Figures, the statistical significance was assessed using a two-tailed paired sample *t*-test. Abbreviations, ACS; Autologous conditioned serum, SP; Sham plasma, IL-1RA; Interleukin-1 Receptor Antagonist, VEGF; Vascular Epithelial Growth Factor, TGF-β1; Transforming Growth Factor beta 1, IL-1β; Interleukin-1 beta, conc.; concentration.

**Figure 2 animals-12-02717-f002:**
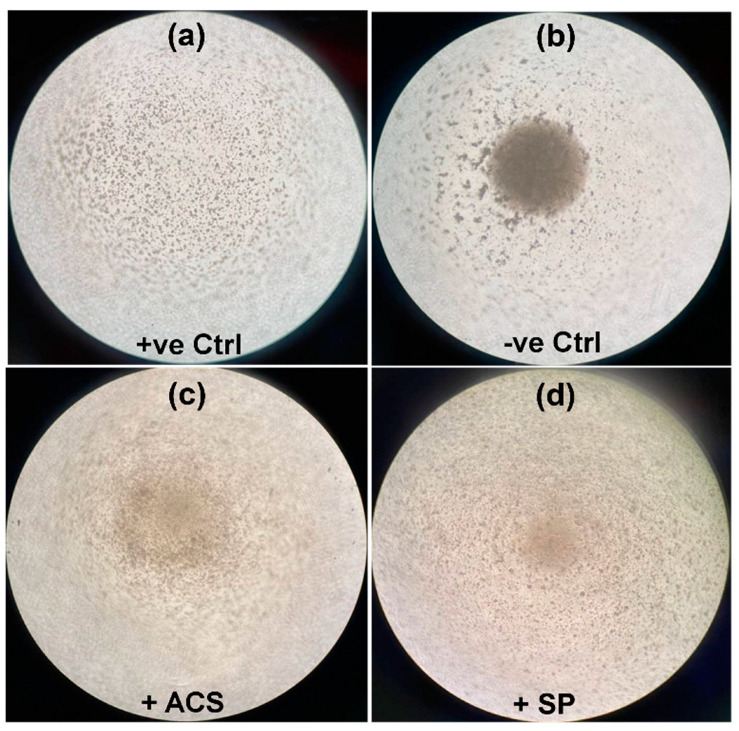
Images representing the cell proliferation condition observed under light microscopy. Representative images of an overview observation on T cells proliferations are shown in the panel. The proliferated cells in (**a**) the positive control well (+ve Ctrl) appears as scattered distribution of cells whereas (**b**) the unstimulated cells (-ve Ctrl) accumulated at the bottom of the U-shape well. Increased clumping of cells indicates no cell proliferation found in (**c**) ACS treated group more than (**d**) SP. The cells were then immunostained and quantified using flow cytometry analysis. All panels were the images captured at 40X magnification under the light microscope. Abbreviations, +ve Ctrl; positive control, -ve Ctrl; negative control, ACS; Autologous conditioned serum, SP; Sham plasma.

**Figure 3 animals-12-02717-f003:**
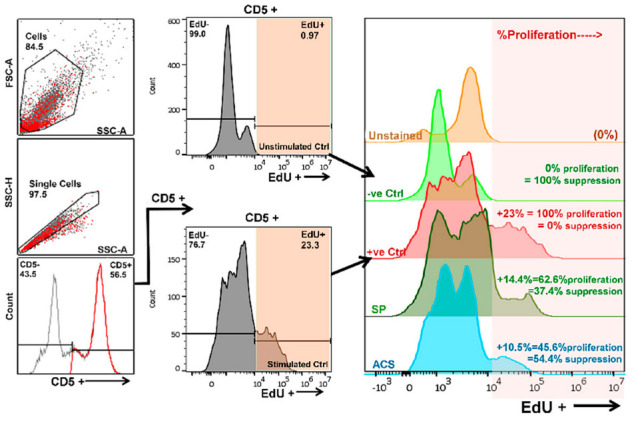
Flow cytometry gating and percentage of suppression calculation. The diagram shows the flow cytometric gating scheme. The EdU^+^ cells in the CD5^+^ population were targeted. For each individual, the suppression percentage was calculated and normalized to their assay controls; unstimulated control (100% suppression) and stimulated control (0% suppression). The image shows the calculation of a representative dog. Abbreviations, CD; Cluster of differentiation, FSC; Forward scatter, SSC; Side scatter, EdU; 5-ethynyl-2′-deoxyuridine, Ctrl; Control, ACS; Autologous conditioned serum, SP; Sham plasma.

**Figure 4 animals-12-02717-f004:**
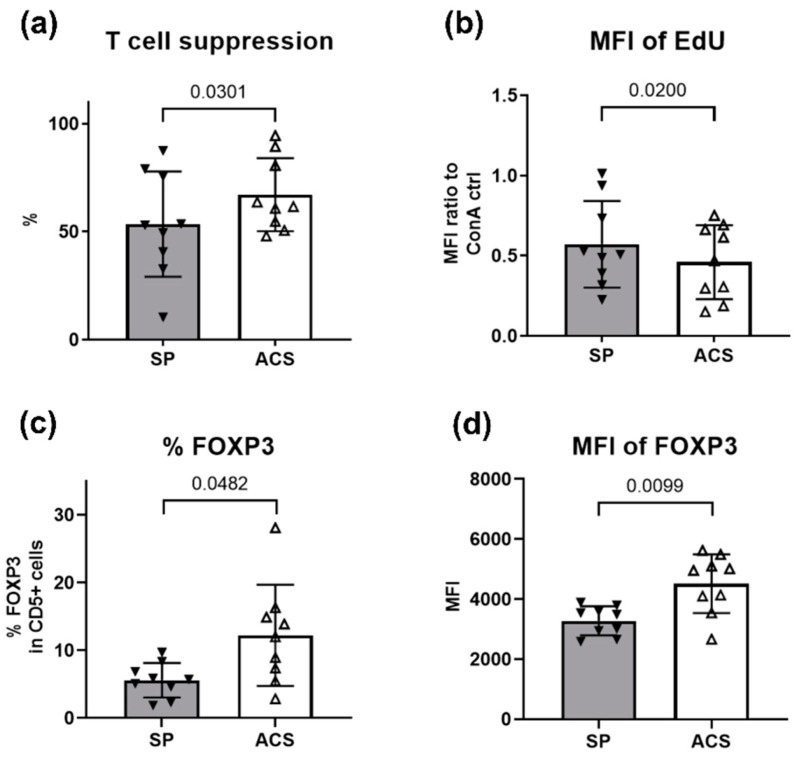
Percentage of T cell proliferation suppression; Bar graphs show the analyses of T cell activity. Error bars depict means with SDs in all panels. (**a**) The percentage of T cell suppression, and (**b**) MFI ratio of EdU in T cells normalized to control are presented. Both ACS and SP groups showed an ability to inhibit T cell proliferation. The ACS group has a greater effect on T cell suppression than SP, and the MFI of EdU-incorporated DNA was lesser in ACS. Evaluation of the FOXP3 population in response to ACS and SP is shown, (**c**) the percentage of FOXP3^+^ T cells and (**d**) MFI of FOXP3 expression. Filled and blank triangles represent the mean results obtained from 10 OA dogs with technical triplicates. For all Figures, statistical significance was assessed using a two-tailed paired sample *t*-test, and *p* values are indicated. Abbreviations, ACS; Autologous conditioned serum, SP; Sham plasma, MFI; Mean fluorescent intensity, EdU; 5-ethynyl-2′-deoxyuridine, ConA; Concanavalin A, CD; Cluster of differentiation, FOXP3; Forkhead box P3.

**Figure 5 animals-12-02717-f005:**
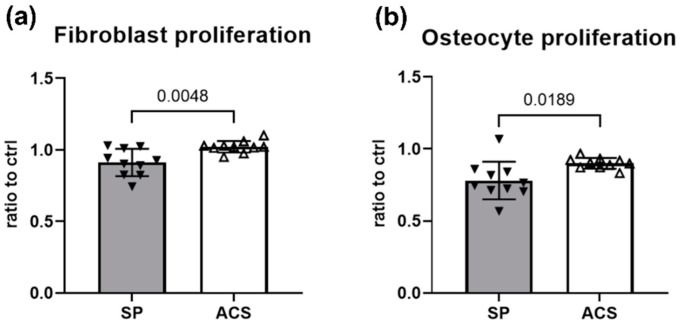
Regenerative property of ACS in promoting fibroblast and osteocyte proliferations. Bar graphs show the cell proliferation ratios using the MTS assay. Error bars depict means with SDs in all panels. The ACS effects on (**a**) fibroblast and (**b**) osteocyte proliferations were shown. *Y*-axis indicates the normalized values ratio to control medium culture. Statistical differences were calculated using a two-tailed paired sample *t*-test. Filled and blank triangles are representatives of the mean results obtained from 10 OA dogs, three independent experiments for each cell line and technical triplicates for each round. Abbreviations, ACS; Autologous conditioned serum, SP; Sham plasma, Ctrl; Control (cells grown in the standard complete media recommended for each type of cell line).

**Figure 6 animals-12-02717-f006:**
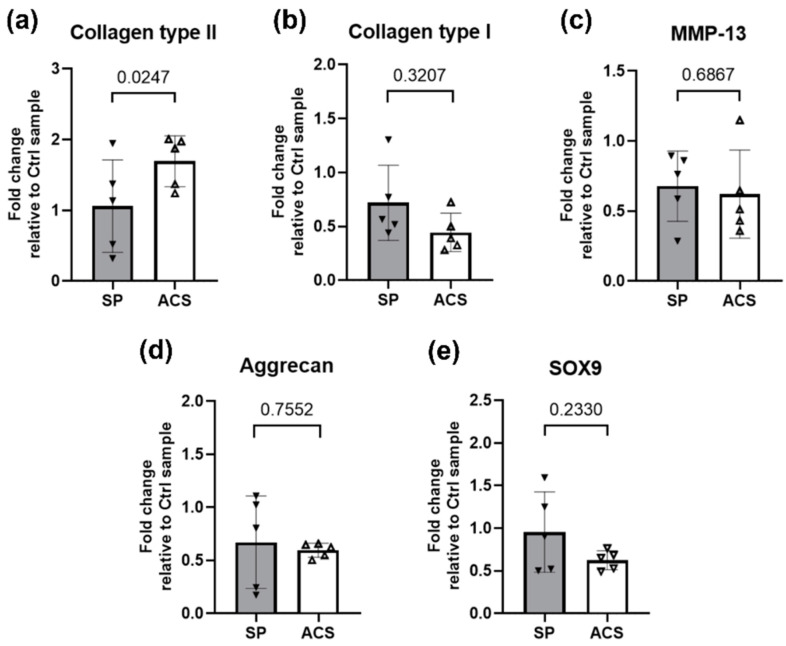
Effect of ACS on gene expression of cartilage matrix protein. Bar graphs show the average fold change in mRNA expressions calculated using 2^−ΔΔCT^ method. Error bars depict means with SDs in all panels. Respectively, the fold change in the expression of (**a**) Collagen type II, (**b**) Collagen type I, (**c**) MMP-13, (**d**) Aggrecan, and (**e**) SOX9 are reported. Statistical differences were calculated using a two-tailed paired sample *t*-test. Filled and blank triangles are representatives of the mean results obtained from five individuals, two independent experiments using primary chondrocytes from two different, unrelated donors. Abbreviations, ACS; Autologous conditioned serum, SP; Sham plasma, Ctrl; Control, MMP-13; Matrix metallopeptidase 13, SOX9; SRY-Box Transcription Factor 9.

**Table 1 animals-12-02717-t001:** Demographic data of study group.

	Breed	Age	Gender	Duration of OA	Affected Joint (s)
1	Golden Retriever	8 Y, 6 M	Male	1 year	bilateral hips
2	Golden Retriever	4 Y, 10 M	Female	4 weeks	stifle
3	Labrador Retriever	4 Y, 7 M	Female	8 weeks	stifle
4	Labrador Retriever	3 Y, 4 M	Male	4 weeks	bilateral stifles
5	Crossbreed	8 Y, 7 M	Male	5 weeks	bilateral stifles
6	American Pitbull	3 Y, 4 M	Female	3 weeks	stifle
7	Golden Retriever	2 Y, 2 M	Female	3 weeks	bilateral stifles
8	Siberian Husky	8 Y, 2 M	Male	3 weeks	stifle
9	German Shepherd	8 Y, 7 M	Male	4 weeks	elbow
10	Labrador Retriever	6 Y, 8 M	Female	1 year	bilateral stifles

Abbreviation, Y; Year, M; Month, Note: Duration of OA is counted after undergone the standard treatment as mentioned in Methods.

## Data Availability

The data presented in this study are available within the article. Raw data supporting this study are available from the corresponding author.

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
