# Peer review of "In Vitro Anti-Inflammatory and Regenerative Effects of Autologous Conditioned Serum from Dogs with Osteoarthritis"

_animals, 2022, doi:10.3390/ani12192717_

Round 1

Reviewer 1 Report

Dear authors, 

I find your work really interesting. As you mention, you deal with a current and not too much studied topic, so your work could be useful in the improvement of our dog's life quality and welfare. 

I consider that the article is properly structured and written, and attached tables and images help in its understanding. 

Additionally, you should consider these suggestions that, in my opinion, will improve the quality of the article: 

·      Line 137: describe the incubation process, for example, where was the incubation performed? What temperature was used? 

·      Figures should be mentioned in the main text in order of appearance. In the line 170, figure 3 is mentioned, however, it is the first figure, so the name must be changed. Moreover, figure 1, that can be found in line 248, is not cited in the main text. 

·      Please use the abbreviations previously described in the main text in all the figures.

·      Line 308 and 309: different font type

·      With regards to sample size, how have you determined a N of 10 animals? 

Reviewer 2 Report

The article is well writtent and the methods described are consistent and detailed. Results and discussion are also sound and well described.

The title suggests that the study describes a clinical effect of ACS. However, all assays are in vitro trials with plasma from OA dogs. To avoid misinterpretations. I would suggest to include "in vitro" or "ex vivo" at the beggining of the title.

Line 131. How sedated dogs are given free access to water? I suggest to remove this first sentence?

The conclusions section is a little too long and vague. It repeats some of the results. Here, some specific 2-3 concrete conclusions from the results of the study shuld be briefly outlined. 

minor spelling:

line 84 very should read vary

line 356 dogs in plural

Reviewer 3 Report

The manuscript deals with the potency of ACS obtained from OA dogs. This is important and interesting, and the authors choose a wide range of assays to investigate ACS potency. However, there are some shortcomings that need to be clarified. Either the description of what has been done and calculated is not sufficient, or some of the assays do not support the conclusions drawn (see below). Furthermore, language/ writing in general must be improved. Some sentences appear incomplete.

Mediators:
- Please specifiy more details of your method. ELISA in blood products can be tricky. Dilutions used for each measurement should be given.
- Which TGFb was measured?
- Please do a more detailed comparison to cytokine levels found in healthy dogs (e.g. Sawyere et al.)- IL1-Ra levels in that former study were much higher, which needs discussion.

T cell proliferation assay:
- Did you actually compare ConA stimulation vs ACS or plasma, and then consider it as suppressive effect when proliferation was less with ACS and plasma? Did the ACS and plasma groups contain ConA as well? If yes, make it clear. If no- the conclusion is invalid, because the mere fact the ACS or plasma stimulate the T cells less than ConA does not mean they suppress them (actually they would still stimulate them as compared to negative control).
- The Click-iT is not an "immunostaining" (see methods) and the control is not an "isotype" control, I think (maybe "unstained"?) (see figure 3).

Cell proliferation:
Please specify how the "ratio" was calculated. As the ACS and plasma supplemented groups have a value below 1, it seems more like both did not support proliferation? Rather, ACS had "less negative impact" on it?

Chondrocyte gene expression:
- It is too vague to conclude that ECM/ Col2 "production" is enhanced (you measured the gene expression only). Change the wording.

Reviewer 4 Report

IN my opinion it would be important to clarify in the title that the study was conducted in vitro using ACS from dogs.

IN some parts the text was hard to read (long sentences). May be improved to make more easy to read. 
